# REAL-TIME NEURAL-BASED INPUT METHOD

## ABSTRACT

The input method is an essential service on every mobile and desktop devices that provides text suggestions. It converts sequential keyboard inputs to the characters in its target language, which is indispensable for Japanese and Chinese users. Due to critical resource constraints and limited network bandwidth of the target devices, applying neural models to input method is not well explored. In this work, we apply a LSTM-based language model to input method and evaluate its performance for both prediction and conversion tasks with Japanese BCCWJ corpus. We articulate the bottleneck to be the slow softmax computation during conversion. To solve the issue, we propose incremental softmax approximation approach, which computes softmax with a selected subset vocabulary and fix the stale probabilities when the vocabulary is updated in future steps. We refer to this method as incremental selective softmax. The results show a two order speedup for the softmax computation when converting Japanese input sequences with a large vocabulary, reaching real-time speed on commodity CPU. We also exploit the model compressing potential to achieve a 92% model size reduction without losing accuracy.

## 1 INTRODUCTION

Typing is one of the core activities users interact with devices. The input method is an intelligent service in operating systems that provides text suggestions to boost typing efficiency. With a large user base, commercial input products have helped users save trillions of key presses. Being able to provide an input method service in high quality makes a huge business impact.

Among various features an input method provides, we consider two essential text suggestion tasks in this work: next word prediction task and conversion task. Next word prediction task relies on a language model, which gives the probability of the next word $p(w_t|w_{1:t-1})$. Usually, top 3 or 5 candidates are shown to users. Conversion task, however, is not a common scenario for Latin users. Latin users can type directly with a keyboard, but there are thousands of characters in the Chinese and Japanese language. Conversion task takes a key sequence and converts it to target word sequence that best matches users' original intentions. It is a sequence decoding task similar to machine translation.

Conventionally, an n-gram language model is used to solve these two tasks (Stolcke, 2002). However, due to the exponential increase in model size , the n-gram model can only support up to tri-gram. The decoding results often fail to have global consistency as the tri-gram language model is not capable of considering distant context. Also, n-gram models are often pruned to meet model size requirement with a loss in accuracy. In contrast, neural-based language models are proven to solve such issues more efficiently(Bengio et al., 2003; Mikolov et al., 2010; Jozefowicz et al., 2016).

Although neural-based language model has achieved enormous success, conventionally, it is not considered feasible for input method task. One major reason is that the input method has to run interactively at user devices with critical constraints of resources and run-time speed. This is different from speech recognition or machine translation services that are normally served from online servers. Input applications have to sacrifice accuracy for execution speed and model size.

In this work, we enhance a neural language model tailored for input method task to meet both runtime speed and model size requirement. Our baseline model is composed of a LSTM-based language model (Hochreiter & Schmidhuber, 1997) with a Viterbi decoder (Forney, 1973) as Figure 1 shows.

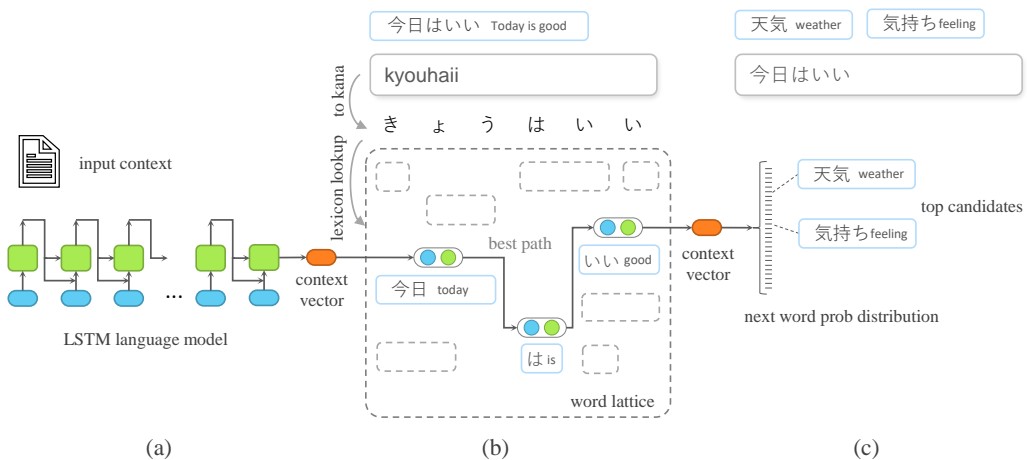

Figure 1: Illustration of the proposed neural-based input method. (a) Input context. (b) Conversion with LSTM LM and Viterbi decoder. (c) Word prediction.

It improves the performance for both conversion and next word prediction task and is capable of capturing distant context.

For languages like Japanese and Chinese, a large vocabulary is required in the conversion task. we articulate the bottleneck of run-time speed as the vocabulary size matrix operation before softmax. We propose an incremental selective softmax approach that builds a subset vocabulary $V_t$ at each decoding step $t$. By only computing softmax over $V_t$, the cost of softmax significantly drops since $|V_t|$ is usually a small number that is less than $1\%$ of the original vocabulary size. We evaluate the speedup comparing to other softmax optimization. To further handle the model disk size and memory size, we apply k-means based quantization algorithm and analyze its performance with various compression ratio.

We summarize the main contributions of this paper as follows:

1) We empirically evaluate using a LSTM-based language model to address input method challenges and demonstrate its performance for a complex language with large vocabulary size.

2) We propose an effective approach to accelerate the inference time with incremental selective softmax. By avoiding computing the full softmax, our approach achieves two orders inference speed boost without negatively impact the accuracy of conversion.

3) To let the neural language model with a large vocabulary fit into the execution environment on consumer devices, we compress the model with quantization techniques, achieving a 92% reduction on model size.

## 2 NEURAL-BASED INPUT METHOD

The neural-based approach we propose for input method is illustrated in Figure 1. We use LSTM language model for both the conversion and prediction tasks. A noticeable difference from the n-gram approach is the use of a latent vector to capture input context. It is the hidden vector from the last recurrent unit of the previous sequence. In a real scenario, users don't type the full sentence in one effort. Instead, they type and convert in fragments, where each fragment contains a few words. It is important to carry the context of the previously determined results to current conversion task.

Conversion task is described in Figure 1(b). As the user starts to type in an editor control, the input method first converts the alphabet user typed into syllables. Take Japanese as an example, Japanese Kana, also called "fifty sounds", can be converted from the alphabet with fixed rules. In lexicon, Japanese words are indexed by Kana characters with an efficient structure such as Trie. A word lattice is then built with all possible words from a Kana sequence by lexicon lookup. To find the best combination of words aligned with the Kana sequence, we use a Viterbi decoder to decode with

a beam size $S$. LSTM language model is used here to evaluate path probabilities. The best path is then shown to the user as the conversion candidate.

Once the user selects the conversion candidate, last hidden vector of the best path becomes the new context vector. We compute the softmax based on the context vector and provides a probability distribution of the next words as shown in Figure 1(c).

Although several commercial input products have already tried to apply neural models[1], they mainly aim to solve the prediction task for Latin languages. Conversion task has a different scale of computational cost and is not yet tackled.

## 3  MODEL ACCELERATION

### 3.1  BACKGROUND

Comparing to a prediction task, the conversion task is even more costly to compute. We further articulate three issues that make the conversion task difficult to solve: 1) For Japanese or Chinese input method, a much larger vocabulary size is required, which is usually several times larger than English. 2) During decoding, we have to evaluate probabilities for all paths with a beam size $S$. 3) Most importantly, there are no natural word boundaries in Chinese or Japanese, such as a space character. The path evaluation has to run each time a new key input arrives, instead of when space is inserted. With all above constraints, the conversion task has to compute $S \times |V| \times |X|$ steps of LSTM computation for one input sequence, where $|V|$ is the vocabulary size, $|X|$ is the sequence length.

Heavy computation cost of LSTM model comes from two operations: the matrix operations inside LSTM cell and the matrix projection at softmax. In the case of a LSTM model with a vocabulary size of 100K, a hidden size of 512, and an embedding size of 256, estimated by simple matrix multiplication, the number of multiplication operations for LSTM cell is about 1.5M, while the softmax has 50M operations. In practice, the softmax occupies 97% of the total computation cost, which is the bottleneck for the conversion task with a large vocabulary size.

### 3.2  INCREMENTAL SELECTIVE SOFTMAX

To solve the challenges, we propose an incremental selective softmax approach to obtain correct probabilities for paths in the lattice. Given a partial observed character input sequence $(x_1, ..., x_t)$, we search for a set of words in the dictionary that match any suffix of the observation:

$$D_t = \bigcup_{i=1}^{t} \text{match}(x_i, ..., x_t), \tag{1}$$

where the function $\text{match}(\cdot)$ returns all lexicon items matching the partial sequence. For example, given the observation "ha ru ka", the lexicon set $D_t$ contains all words matching "ha ru ka" or "ru ka" or "ka". With the lexicon set in each step, we can construct a subset vocabulary $V_t$ that covers all possible output words until step $t$ as:

$$V_t = \bigcup_{i=1}^{t} D_i. \tag{2}$$

In this work, we refer to $V_t$ as the lattice vocabulary. In contrast to the beam search in machine translation, we have to rank paths that contain different number of words. If we directly compute path probabilities on $V_t$ with a language model, as the denominator of the softmax becomes smaller, all words will have higher probabilities. Therefore, the paths containing more words will have advantage due to the incorrect probabilities. As a result, the paths can be ranked wrongly.

To circumvent the probability issue, we also sample a subset vocabulary from the full vocabulary, denoted by $\tilde{V}$. We use $V_t$ jointly with $\tilde{V}$ to make the word probabilities closer to their original probabilities, thus minimizing the impact of exposure bias.

---

[1]SwiftKey, Fleksy, etc.

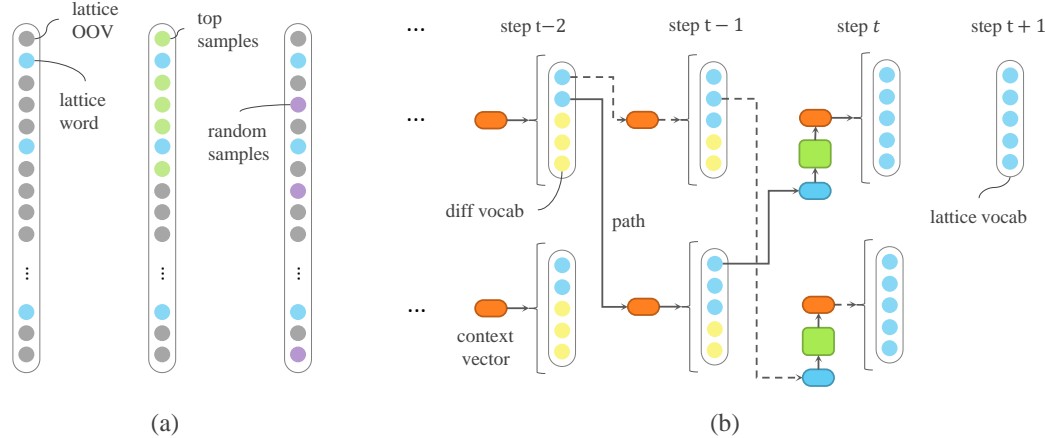

(a)                                                                    (b)

Figure 2: Illustration of incremental selective softmax: (a) Vocabulary sampling strategies (b) Incrementally correct path probabilities with new lattice vocabulary.

As shown in Figure 2(a), we experiment two sampling methods. The first one samples mostly frequent words from the vocabulary, whereas the second one uniformly samples from the full vocabulary. We found the first sampling method (top sampling) delivers the best performance. In practice, both $V_t$ and $\tilde{V}$ have less than $1\%$ of the words in the original vocabulary.

Our proposed softmax approximation technique has two passes. When a new input character $x_{t+1}$ arrives, we merge the current vocabulary $V_t$ with new words from the lexicon set $D_{t+1}$, forming a new vocabulary $V_{t+1}$. Then for each path in the lattice, we compute the language model probability at step $t + 1$ with the joint vocabulary $V_{t+1} \cup \tilde{V}$ as:

$$P(y_t = i|h_t) = \frac{\exp(h_t^\top W_i)}{\sum\limits_{j \in V_{t+1} \cup \tilde{V}} \exp(h_t^\top W_j)}, \tag{3}$$

where $h_t$ is the LSTM hidden state in step $t$. In this pass, we only compute the word probabilities for top paths in batch with a beam size $S$.

Now we computed the latest word probabilities using the lattice vocabulary $V_{t+1}$. Unfortunately, as previous word probabilities are computed on old vocabularies, they are not comparable with the new ones. Therefore, in the second pass, we correct the stale word probabilities to let them have the same denominator as the ones computed on the latest lattice vocabulary.

In principle, we can correct an old probability in step $k$ by adding the logits of missing vocabulary to the denominator as:

$$P(y_k = i|h_k) = \frac{\exp(h_k^\top W_i)}{\sum\limits_{j \in V_{k+1} \cup \tilde{V}} \exp(h_k^\top W_j) + \sum\limits_{j \in V_{t+1}, j \notin V_{k+1} \cup \tilde{V}} \exp(h_k^\top W_j)}. \tag{4}$$

However, in practice, as each path has different missing vocabularies, we compute a union of all missing vocabularies, and then recompute the logits of them in batch. Experiments show that our proposed incremental selective softmax strategy achieves a 76x speedup comparing to the baseline.

## 4    EXPERIMENTS

In this section, we compare the performance of the neural-based approach with conventional approaches. Then, we report the results of applying model acceleration and compression techniques.

## 4.1 DATASET

We use BCCWJ (Balanced Corpus of Contemporary Written Japanese) corpus (Maekawa et al., 2014) for evaluating our model. The corpus is well balanced with various sources of text representing contemporary written Japanese. This corpus contains 5.8M sentences, which are segmented into 127M tokens. In our experiments, all words are further segmented into short unit words. Each word has a format of "display/reading/POS". The reading and POS attributes are attached to tell different usages of the same word. Among the 611K unique words, we choose top 50K frequent ones that cover 97.3% of token appearances as an appropriate vocabulary size for the input method task. The words in the vocabulary are ranked with frequency. Most frequent words are at the top.

## 4.2 EXPERIMENT SETTINGS

The BCCWJ dataset is split into training, valid, and test set. The ratio is 70%, 20%, and 10% respectively. We randomly sample 2000 sentences from the test set for evaluating conversion accuracy. For input method task, we evaluate the conversion accuracy using a Viterbi decoder with a beam size of 10. In Japanese, there are often more than one correct conversion results. For instance, a verb may have two acceptable styles, one in original Japanese Kana, the other in Chinese characters. To better evaluate the model performance, we also report the top-10 accuracy in addition to top-1 accuracy.

We implement the model using TensorFlow[2]. We use a batch size of 384, and dropout with a drop rate of 0.9. Adam optimizer is used with a fixed learning rate of 0.001. The hyper-parameters are shared for all experiments.

A replica of the same model is written in numpy to work with a Viterbi decoder in python. It uses the weights learned with TensorFlow model. The inference performance is measure with numpy on a single Intel E5 CPU. We also apply the underline BLAS library to accelerate matrix operation.

## 4.3 EVALUATION OF NEURAL-BASED INPUT METHOD

We first compared the neural model performance with a conventional n-gram model. We calculate the perplexity of the n-gram model with SRILM package (Stolcke, 2002). We choose modified Kneser Ney (Kneser & Ney, 1995; James, 2000) as the smoothing algorithm when learning the n-gram model. The learned language model is plugged into our input method pipeline for evaluating the sentence conversation accuracy. The prediction accuracy is not provided as it is directly reflected by model perplexity.

The LSTM baseline model has a standard architecture. The embedding size is 256, and the hidden size is 256. The size of LSTM cells is selected empirically on a validation dataset. In practice, using a network size bigger than 256 cannot gain significant improvement on perplexity. In all following experiments, we bind the input embedding and output embedding according to Press & Wolf (2017). The idea is proven to save space and almost loss-less. We treat it as the baseline model in following experiments.

| Models | pp | top1 hit % | top10 hit % |
|---|---|---|---|
| uni-gram | 833.55 | 26.95 | 45.85 |
| bi-gram KN | 99.30 | 51.15 | 78.10 |
| tri-gram KN | 68.11 | 55.60 | 79.65 |
| LSTM baseline | **41.39** | **61.20** | **88.30** |

Table 1: Performance comparison of baseline LSTM model with conventional n-gram model.

As shown in Table 1, the LSTM baseline achieved a significant improvement on perplexity, comparing to conventional n-gram based models. The top-1 and top-10 path conversion accuracy were increased by 5.6% and 8.65% respectively comparing to tri-gram KN. In real products, bi-gram is often used for decoding while tri-gram is only used for re-ranking the best paths. Please note that in this evaluation, we didn't apply any pruning for n-gram. The n-gram model takes over 1GB storage size.

---

[2]We implement a standard LSTM cell to avoid any unexpected customization from published version

## 4.4 EVALUATION OF RUN-TIME SPEED

In this section, we compare the inference speed of various softmax acceleration approaches. We measure the execution speed only for the component that computes the language model probabilities. For neural-based methods, the component includes the LSTM and softmax layer. For the n-gram model, the computation of probability is only a lookup in the hash tables. Other components such as lattice construction are not included as they heavily depend on implementation.

Here, we report the decoding time in each step receiving a key input, because the per-step latency is critical for the real-time requirement. The computation cost for decoding the whole sequence is linear to the number of steps. For comparison, we also report the computation time of the softmax alone.

| Models | Total Time (ms) | Softmax Time (ms) | Top-1 Hit (%) | Top-10 Hit (%) |
|---|---|---|---|---|
| tri-gram Kneser-Ney | 0.0025 | - | 55.6 | 79.6 |
| LSTM baseline w/o batch | 526 | 513 | 61.2 | 88.3 |
| LSTM baseline w/ batch | 87 | 84 | 61.2 | 88.3 |
| Char LSTM | 63 | 21 | 53.2 | 79.8 |
| Char LSTM w/ large beam[3] | 152 | 55 | 54.4 | 85.2 |
| D-Softmax | 38 | 35 | 60.8 | 86.8 |
| D-Softmax* | 37 | 35 | 60.9 | 87.5 |
| IS-Softmax | 3 | 1 | 58.8 | 86.9 |
| w/ top sampling | **4** | **2** | **60.1** | **88.2** |
| w/ uniform sampling | 4 | 2 | 58.9 | 87.2 |
| non-incremental S-Softmax | 11 | 2 | 58.8 | 86.9 |

Table 2: Model acceleration among different methods with accuracy.

As Table 2 shows, our proposed incremental selective softmax (IS-Softmax) achieves a 76x speedup comparing to the LSTM baseline. The lattice vocabulary in our experiments contains only a few hundred words, while the full vocabulary has 50k words. IS-Softmax only takes 3 ms to handle a new coming key. Such a high processing speed meets the real-time latency requirement.

As a comparison, the non-incremental selective softmax is less efficient since it recalculates from the beginning in each step. Therefore, the majority of its cost comes from computing LSTM cells. We also evaluate various sampling methods with IS-Softmax. We find top sampling help close the accuracy gap between the baseline and softmax approximations. In our experiments, the number of samples used in top sampling and uniform sampling methods are both 400.

Differentiated softmax (D-Softmax) (Chen et al., 2016) and its variation (D-softmax*) (Joulin et al., 2017) can also reduce the amount of softmax computation by over a half depends on the segmentation strategy. However, since the vocabulary size is still large, the room for speedup is limited. For character-based LSTM models, we use a hidden size of 1024 and embedding size of 512. Since there is no direct mapping between a single Kana and a single Chinese character, we use the word lattice and evaluate the path probability by character-based LSTM. It reduces the vocabulary size from 50K to 3717 in this experiment. However, the amount of vocabulary is still non-trivial. Furthermore, the integration of character-based models is not as efficient as word-based models in this task. Consequently, we have to use a large beam size to improve accuracy.

We also confirmed that batching is critical for acceleration matrix operations even on CPU. The LSTM computation without batching is almost 7x slower with a beam size 10. Therefore, we batch all the softmax computations when fixing the vocabulary in the second pass of IS-Softmax to achieve the best speed.

---

[3] The large beam size is set to 50 in our experiments.

## 4.5 Evaluation of Model Compression

Due to the limited resource on user devices, the model for the input method has to be extremely compact for both storage and memory footprint. For the mobile platforms, the distribution package that contains the neural model shall have a size small than 10 MB in order to handle various network environments. For commercial products, a large model size directly hinders the adoption of neural models even a higher accuracy is achieved.

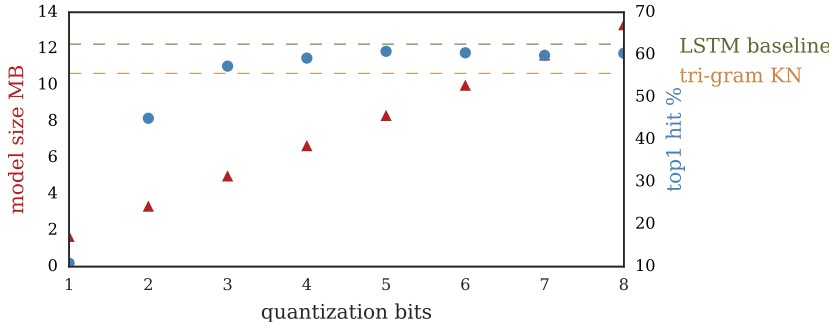

Figure 3: Model compression with k-means clusters from $2^1$ to $2^8$.

In our experiments, we implement a k-means clustering algorithm (Han et al., 2016) to compress our neural language model. Instead of applying clustering method to all the weights of the LSTM model, we generate code and codebook for each trained matrix. As Figure 3 shows, the quantization method using codes of 8 bits works well without degrading the conversion accuracy. Even with very low bits such as 3 and 4, the conversion accuracy is still higher than the n-gram baseline. The model size for n-gram has over than 1GB storage footprint. In practice, pruning the n-gram model inevitably causes a loss in accuracy. The neural language model shows a big advantage over n-gram in terms of model compression. By applying 5-bit quantization, together with the tie embedding approach, we reduce 92% of model size without noticeable accuracy loss.

## 5 Qualitative Analysis

Among the sentences that n-gram cannot simply decode, we select a few examples to perform a qualitative study from the user experience point of view. Figure 4 shows the decoded results from the n-gram and LSTM-based language models. The text in red shows the wrong conversion results, and the text in blue is the correct results. According to human judgment, the most important hint words for a correct conversion are set to bold font. To better align with the Japanese text, the English translation is not in its original order.

We observe that the LSTM-based language model is capable of capturing distant context and producing correct results that are difficult for a statistical language model to predict. In case 1, the writer uses parallel structure to describe the core difference between two famous universities in Japan. Such long tail sentence is very creative and doesn't appear twice in the corpus. Also, the middle word is separated by parenthesis, which causes trouble for the n-gram model. As word embedding can easily identify the antonyms, the LSTM-based model successfully predicts the correct output in such cases. Case 2 and 3 show that distant trigger can be captured by the neural model. Even when the trigger word is in a distant context, the LSTM-based model still understands the topic and selects the correct candidate as case 4 and 5 demonstrate.

We also found that when two correct candidates exist, n-gram models tend to keep only one of them in the lattice. In contrast, the LSTM-based model keeps the variety in the lattice and thus resulting in a better top 10 hit rate. We argue that the word embeddings for such interchangeable words are very close in the vector space after training. As a result, they retain similar probabilities during the decoding process.

| | | |
|---|---|---|
| 1 | n-gram | 論理（具体）の東大、発送（中小）の兄弟
**Logical (concrete) University of Tokyo**, delivery (medium small) brother |
| | LSTM baseline | 論理（具体）の東大、発想（抽象）の京大
**Logical (concrete) University of Tokyo**, creative (abstract) University of Kyoto |
| 2 | n-gram | 昔は駐車だった**抗生物質**
In the past parking **antibiotics** |
| | LSTM baseline | 昔は注射だった**抗生物質**
In the past injected **antibiotics** |
| 3 | n-gram | **自民党**だけとなりますと非常に偏っ体験
**Liberal Democratic Party** only about very biased experience |
| | LSTM baseline | **自民党**だけとなりますと非常に偏った意見
**Liberal Democratic Party** only about very biased opinion |
| 4 | n-gram | 子供二人を幼くして**亡く**し、百二十二歳で妻も高い
Children as child **lost life**, at the age of 122 wife also tall |
| | LSTM baseline | 子供２人を幼くして**亡く**し、百二十二歳で妻も他界
Children as child **lost life**, at the age of 122 wife also died |
| 5 | n-gram | **飛行場**へ行ってみると、予約を入れてあったのに、何かの都合でその日はに気飛ぶはずのものが一気しか飛ばなかったのである
**Airport** went to, reservation had made, for some reason at the day Ki didn't flew only one stroke flew. |
| | LSTM baseline | **飛行場**へ行ってみると、予約を入れてあったのに、何かの都合でその日は二機飛ぶはずのものが一機しか飛ばなかったのである
**Airport** went to, reservation had made, for some reason at the day two aircraft didn't flew only one aircraft flew. |

Figure 4: Case study between conversion results of LSTM-based language model and n-gram model.

# 6 RELATED WORKS

Applying a neural-based model for input method is studied in a few previous works. Chen et al. (2015) proposed a MLP architecture for Chinese Pinyin input method. Due to the huge computation cost of matrix operations, the neural model is only applied to rank the best paths decoded with n-gram language model. Though not in literature, we have found implementation of RNN-based input method[4]. The decoding results on BCCWJ corpus show promising accuracy improvement comparing to the n-gram model. Huang et al. (2018) treat the Chinese input conversion as machine translation and apply seq2seq with attention model. The conversion model and a few other features are served as cloud services. Our work focus on enabling neural-based models on devices especially with limited computation resources.

While there are various choices for the neural-based model, we choose a single layer LSTM model (Hochreiter & Schmidhuber, 1997) primarily due to the run-time speed consideration. For other DNN models, we argue that seq2seq like models (Cho et al., 2014) are not feasible to input method task because in each step the decoder needs to compute from start with a new latent vector. Bi-directional architectures such as Bi-LSTM (Huang et al., 2015) normally requires full input sequence beforehand, it cannot compute incrementally. Character-based RNN (Mikolov et al., 2010) is a good alternative for word-based models. It has a smaller vocabulary size. However, for Chinese and Japanese, there are still thousands of characters. Also, it doesn't predict the next word directly.

To reduce the cost of softmax computation, Chen et al. (2016) proposed a differentiated softmax (D-Softmax), which computes the logits with weight matrices differ in size depending on word frequency. It argues that less frequent words require fewer softmax parameters compared to frequently used words. Joulin et al. (2017) is a variation of the D-softmax with more efficient weight segmentation. Together with the tie embedding approach (Press & Wolf, 2017), the cost of the differentiated softmax becomes $\sum_{i=0}^{n} |V_i| \times E_i$, where each $E_i$ is smaller value than its original size. Although the embedding optimization approach can reduce softmax cost, the amount of reduction is limited. It is still proportional to the reduced embedding size.

---

[4]Yoh Okuno. Neural ime: Neural input method engine. https://github.com/yohokuno/neural_ime, 2016

Another direction of reducing softmax cost comes from vocabulary manipulation. In machine translation, given a source sentence, the vocabulary of target words can be limited before translation. It is possible to calculate the softmax on a subset of the full vocabulary (Jean et al., 2015; Mi et al., 2016). However, for the input method, the conversion task takes user input incrementally. Our proposed incremental selective softmax can work without knowing full vocabulary beforehand. It is generally applicable for incremental sequence decoding task with a large vocabulary.

## 7 CONCLUSION

Input method plays an important role in improving typing efficiency and user experiences. It has to work on various devices with minimized latency. We study the key challenges to apply neural-based input method to real commodity devices. The proposed incremental selective softmax significantly reduces cost in the softmax computation without losing accuracy. We also demonstrate that the neural-based models can be highly compressed comparing to the conventional n-gram language models. Our proposed method sets a strong baseline in real-time input method. More importantly, as the computation has a low latency, the model is production-ready to be used on real devices.

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
