# OpenReview forum: "Real-time Neural-based Input Method"
_ICLR.cc/2019/Conference_

### Official Review · AnonReviewer1 · 2018-11-01
**Not enough novelty or significance, unsatisfactory experimental evaluations.**

**Rating:** 3
**Confidence:** 3

**Review:**

Summary: Authors proposed a model for input method for mobile or desktop devices. The goal is to convert the input sequence (from one language to another) or predict the next word. Their model is based on an LSTM with modified softmax activation function that is adjustable for large vocabulary sizes. They showed experimental results on Japanese BCCWJ data set.

Clarity: Paper is well-written and well-organized. Notions and methods are clearly expressed.

Originality: This paper builds on an LSTM model without enough work or idea to show novelty.

Significance: It is below average. Using LSTM is a well-known method for these types of tasks in the literature. Incremental selective softmax is potentially a good approach, however, this work lacks showing significant improvement. The experiments are limited and are done only on one data set.

More detailed comments:

- My concerns about this work are both on modeling aspects and experiments. Authors mainly focus on highlighting the benefits comparing to n-gram models, and briefly discuss the ongoing developments in neural based models. For example sequential modelings using RNN's have shown promising results in capturing long-term dependencies [1]. Unfortunately authors did not include any discussion on how their approach would compare to that framework nor did they present any experimental comparisons to them.

- Although mentioned briefly in the introduction and related work sections, no analytical or experimental comparisons are made to machine translation approaches when their work is closely related to it. I strongly suggest that authors compare their experimental results to some of benchmarks in neural based machine translation discussed in the related works.

- In the incremental selection softmax, they use "match" to return all lexicon items matching the partial sequence. How is this done and what are the effects of it on the computational time of the algorithm? Also, It is not clear how authors correct old probabilities in IS softmax step. As mentioned, they add logits of missing vocabulary to the denominators, how do they keep the properties of softmax so that it sums up to 1? And later in the discussion authors mentioned that in practice they compute union of all missing vocabularies, it is not clear how this is done since the advantage of using IS softmax is expressed to be incremental increasing.

[1] A.B. Dieng, C. Wang, J. Gao and J. Paisley. TopicRNN: A recurrent neural network with long-range semantic dependency, International Conference on Learning Representations (ICLR), 2017.

---

> ### Author Response · Authors · 2018-11-20
> **Thanks for your review and comments!**
>
> Our work originally focuses on conversion task for Japanese and Chinese input method. As reviewer mentioned, it is a better contribution if the approach can be demonstrated on other classic tasks.  We choose a simple LSTM model as a baseline for our selective softmax for its simplicity. Choosing complex network architecture is against our core goal: real-time speed. We will clarify that LSTM model is not part of our contribution by updating the introduction part and methodology sections.
>
> We feel sorry that the model acceleration section is not clearly written at the submission time and left all reviewers questions. We will update his section with a clear algorithm that answers questions reviewers asked.
>
> For a quick answer right now.
> 1. The "match" is a conventional dictionary or trie lookup that returns all possible candidates. The process itself is a O(n^2) scan called build lattice. It is the same process for n-gram and the neural model compared in the work.
> 2. The logits are stored for each frame and path, in that way, we can add the missing logits and run softmax again to calculate a new distribution.
> 3. For the Japanese conversion task, the lattice is not necessarily well aligned. For example, a candidate word may be long and span a large part of the original sequence. In such case, all the frames within the span of the word need to be updated. That is the time we union the missing vocabularies. In practical, such long word is rare, most of the time, only the recent frames are updated.

---

### Official Review · AnonReviewer3 · 2018-11-06
**insufficient novelty, missing competitive baselines**

**Rating:** 3
**Confidence:** 4

**Review:**

The paper demonstrates the main challenge of using LSTM-based language models for input method in real time is the huge amount of computation in the softmax. The authors present a system to speed up the inference by avoiding computing the full softmax in the Japanese conversion task, where the number of output words can be limited from the mapping of the input sequence through a lexicon. The experiment result is encouraging in that the proposed incremental selective softmax approach significantly reduces latency over the standard inference with the full softmax computation while not hurting accuracy much. The paper also evaluates the effect of quantization for LSTM LM model compression in terms of size and accuracy.

However, there are a few major problems of the paper as follows:

1. The main weakness in the experiment setup is that it misses a few competitive baselines in terms of inference speed, notably hierarchical softmax[1] and self normalization[2]. In the Japanese conversion task in the paper, it only needs to evaluate the scores of limited output words that are given from the mapping of the input sequence through the lexicon. This is exactly like the rescoring setup in speech recognition and machine translation, where self-normalization is typically used for efficient inference to avoid computing the expensive softmax normalization term [2,3]. Assuming the number of selected output words is K and the entire vocabulary size is V, then the time complexity is O(K logV) for the hierarchical softmax, O(K) for self normalization, but O(V) for all the baselines in the paper. Self normalization is simple to implement and works well in practice, while the proposed incremental selective softmax approach in the paper needs an additional step to sample most frequent words to adjust the normalization term. Without showing the self normalization result, I am not convinced that the proposed approach is better and needed.

[1] F. Morin and Y. Bengio. "Hierarchical Probabilistic Neural Network Language Model," in Proc. of AISTATS, 2005,
[2] J. Devlin et al., "Fast and Robust Neural Network Joint Models for Statistical Machine Translation," in Proc. of ACL, 2014.
[3] Y. Shi, W. Zhang, M. Cai and J. Liu, "VARIANCE REGULARIZATION OF RNNLM FOR SPEECH RECOGNITION," in Proc. of ICASSP, 2014.

2. The proposed approach would only be useful in speeding up the conversion task, but not applicable to the prediction task where it needs to evaluate all words and choose the top hypotheses. Also how is the latency of the prediction task compared to conversion task? Please also add it to the experiment result.

3. The idea of using quantization for neural network model compression is not novel (even for language model), although it is listed as one of the main contributions in Section 1.

So in general, I think the paper is insufficient in novelty and missing competitive baselines.

Some specific comments:
4. Figure 2(b) is not clear what it means, and not referenced anywhere in the paper.
5. The last 3 lines in Section 3: "as each path has different missing vocabularies": Why is that? The candidates of the output words should only depend on the input sequence and the lexicon, based on Eq(1)(2).
6. It is not clear how to adjust the probability in the second pass of incremental selective softmax. The description "we compute a union of all missing vocabularies, and then recompute the logits of them in batch." is unclear what it means.
7. Section 4.2: "is measure with numpy" -> "is measured with numpy".
8. Section 4.4: It is not clean how "76x speedup" is computed from Table 2 since all the time numbers are rounded. Consider also showing one digit after the decimal point.

---

> ### Author Response · Authors · 2018-11-20
> **Thanks for reviewing and providing very detailed comments!**
>
> Use the original comment index here.
>
> 1. We concern that the hierarchical softmax has lower performance and no gain in runtime speed. But as pointed out by several reviewers, we will add missing baseline for both hierarchical softmax and self-learning.
>
> 2. We would like to focus on the conversion tasks based on the feedbacks. For prediction, there is also some academic work to find top-k candidates. In practice, since prediction only runs every time users finish typing, the delay is not a major issue comparing to the conversion task.
>
> 3. We will remove it from the contribution and move it to experiment.  It shows the results to demonstrate that the model is already usable in real products
>
> 4.5.6
>
> Unfortunately, we didn't make the model acceleration section clearly enough at the submission time. We will update his section with a clear algorithm that answers questions.
>
> For the Japanese conversion task, the lattice is not necessarily well aligned. For example, in a new frame, we find all the candidates ending at the frame. A candidate word may be long and span a large part of the original sequence. Candidates may have a different span and probabilities paths over these new frames need to be updated.
>
> 7.8
>
> Thanks for the detailed comment. We will update.

---

### Official Review · AnonReviewer2 · 2018-11-09
**Weak baseline comparisons and insufficient comparison with prior work**

**Rating:** 3
**Confidence:** 3

**Review:**

This paper describes a search space reduction method for neural network based keyboard input methods. The paper discusses two different sampling methods to restrict the vocabulary size during beam search.

Title: The title of the paper is too generic to describe what is actually being done in the paper. Input methods in mobile devices could have also meant speech based input or handwriting based input or swipe based input. It would be very convenient for the readers if the authors use more specific wording in the title to clarify that they are talking about neural network based keyboard typing input.

Comparison with prior work: Neural network based on-device keyboard input is a research topic with a lot of previous contributions and the existing literature survey seems lacking. Further it does not even cover popular techniques for inference speed-up like hierarchical softmax computation. It would be easier for the reader to appreciate the  contributions of this paper if the authors compare and contrast with more relevant prior work.

---

> ### Author Response · Authors · 2018-11-20
> **Thanks for your review!**
>
> It is a good point that the input method is a generic word and has some ambiguity in terms of mobile device, traditional desktop device.  Also, for Asian audiences and English audiences, input method has a slightly different context. We would like to clarify the specific challenge in this work as the conversion takes for input method editor in next version. We will extend the on-device keyboard researches as well.
>
> For hierarchical softmax, the complexity of the softmax is log(|V|). Assume there are K target words in a frame. Hierarchal Softmax approach requires Klog(|V|) times matrix dot operations to find the probability distribution. The proposed method only runs K times matrix dot operations. Also, due to the word binary encoding choices of Hierarchical Softmax, we doubt its gain from both performance and task accuracy point of view. But as most reviewers pointed out, we believe an analytic comparison is necessary. We will add the experiment results.

---

### Meta-Review · Area_Chair1 · 2018-12-02
**Reject**

**Confidence:** 4
**Recommendation:** Reject

**Metareview:**

All reviewers agree in their assessment that this paper is not ready for acceptance into ICLR.